# Tuning the Interactions in Multiresponsive Complex Coacervate-Based Underwater Adhesives

**DOI:** 10.3390/ijms21010100

**Published:** 2019-12-21

**Authors:** Marco Dompé, Francisco J. Cedano-Serrano, Mehdi Vahdati, Ugo Sidoli, Olaf Heckert, Alla Synytska, Dominique Hourdet, Costantino Creton, Jasper van der Gucht, Thomas Kodger, Marleen Kamperman

**Affiliations:** 1Laboratory of Physical Chemistry and Soft Matter, Wageningen University & Research, Stippeneng 4, 6708 WE Wageningen, The Netherlands; marco.dompe@wur.nl (M.D.); olaf.heckert@wur.nl (O.H.); jasper.vandergucht@wur.nl (J.v.d.G.); thomas.kodger@wur.nl (T.K.); 2Soft Matter Sciences and Engineering, ESPCI Paris, PSL University, Sorbonne University, CNRS, F-75005 Paris, France; francisco.cedano@espci.fr (F.J.C.-S.); mehdi.vahdati@espci.fr (M.V.); dominique.hourdet@espci.fr (D.H.); costantino.creton@espci.fr (C.C.); 3Leibniz-Institut für Polymerforschung Dresden e.V., Hohe Straße 6, 01069 Dresden, Germany; sidoli@ipfdd.de (U.S.); synytska@ipfdd.de (A.S.); 4Laboratory of Polymer Science, Zernike Institute for Advanced Materials, University of Groningen, Nijenborgh 4, 9747 AG Groningen, The Netherlands

**Keywords:** complex coacervate, poly(*N*-isopropylacrylamide), polyelectrolytes, underwater adhesion, environmentally-triggered setting process, LCST, non-covalent interactions, bioinspired materials

## Abstract

In this work, we report the systematic investigation of a multiresponsive complex coacervate-based underwater adhesive, obtained by combining polyelectrolyte domains and thermoresponsive poly(*N*-isopropylacrylamide) (PNIPAM) units. This material exhibits a transition from liquid to solid but, differently from most reactive glues, is completely held together by non-covalent interactions, i.e., electrostatic and hydrophobic. Because the solidification results in a kinetically trapped morphology, the final mechanical properties strongly depend on the preparation conditions and on the surrounding environment. A systematic study is performed to assess the effect of ionic strength and of PNIPAM content on the thermal, rheological and adhesive properties. This study enables the optimization of polymer composition and environmental conditions for this underwater adhesive system. The best performance with a work of adhesion of 6.5 J/m^2^ was found for the complex coacervates prepared at high ionic strength (0.75 M NaCl) and at an optimal PNIPAM content around 30% mol/mol. The high ionic strength enables injectability, while the hydrated PNIPAM domains provide additional dissipation, without softening the material so much that it becomes too weak to resist detaching stress.

## 1. Introduction

The tremendous progress in adhesive technology in the last century has led to the replacement of mechanical fasteners in many industrial fields (automotive, construction) but has not had a huge impact on the biomedical sector, which could gain significantly from the development of surgical glues [1]. For instance, staples and sutures, which are conventionally used for deep tissue bonding, are difficult to apply in small spaces, lead to an extension of operating times and often cause tissue damage [2].

The use of adhesives in the human body is, however, challenging mainly because of the presence of water, which undermines adhesion by weakening the boundary layer or by swelling the adhesive [3,4]. Many natural organisms have solved this challenge and provide us with smart tricks that are worth mimicking [5]: mussels attach strongly to hard surfaces through threads, which are capable of resisting the huge forces exerted by waves [5], and barnacles use secretions to glue calcareous base plates to rocks [6]. Sandcastle worms create a protective shell by connecting sand and shell fragments using a proteinaceous glue [7]: it has been proposed that the delivery and the processing of this adhesive is regulated by a phenomenon known as complex coacervation [8].

Complex coacervates are concentrated phases of oppositely charged polyelectrolytes, having a fluidic character while being phase-separated from water [9]. These properties are beneficial characteristics for an underwater glue: an adhesive should be easily injectable [10] and should not disperse in the surrounding environment [11]. In addition, the typically low interfacial tension of complex coacervates [12] enables them to wet the surface of interest effectively. Finally, limited swelling and shrinking result in good dimensional stability [10].

However, for a proper adhesive performance, additional interactions should be activated after delivery to prevent flow when stress is applied [13]. Natural organisms have engineered a setting mechanism for the fluid complex coacervates in response to a change in the environmental conditions (e.g., exposure to oxygen, change in pH) [14]. Inspired by these systems, material scientists have successfully developed polyelectrolyte-based underwater adhesives whose cohesive properties are activated via covalent crosslinking [15,16,17] and/or by an external trigger (pH [17,18,19], ionic strength [10,20], solvent [21]).

In our previous work [22,23], we have developed a system that combines the presence of polyelectrolyte domains and thermoresponsive units in a graft copolymer architecture: the starting material is obtained by mixing aqueous solutions of oppositely charged polymers having polyelectrolyte backbones (poly(acrylic acid) (PAA) as polyanion and poly(*N*,*N*-dimethylaminopropyl acrylamide) (PDMAPAA) as polycation), grafted with poly(*N*-isopropylacrylamide) (PNIPAM) side chains (see Figure 1). PNIPAM is a well-known thermoresponsive polymer which phase separates from water when the temperature is raised above the so called lower critical solution temperature (LCST) [24], which is generally between 32 °C and 36 °C in demineralized water.

The resulting complex coacervate is able to undergo a liquid-to-solid transition when applying a temperature [22] and/or an ionic strength [23] switch. The promising work of adhesion (*W_adh_*) obtained in physiological conditions makes this material a potential candidate for applications as a surgical glue. The novelty and the beauty of this design lies in the presence of different non-covalent interactions in the same system, which make the adhesive highly tuneable and multiresponsive. Strong bonds, responsible for imparting elasticity to the material, coexist with weaker bonds, that can break and re-form, dissipating energy and providing toughness [25].

In this paper, we present a full report on the effects of salt concentration, temperature and PNIPAM content on the final rheological and adhesive properties of the complex coacervates. This work sheds new light on the correlation between the preparation conditions and the final material properties, enabling the optimization of the system for a specific application, in this case, underwater adhesion in physiological conditions. 

## 2. Results and Discussion

### 2.1. Material Analysis

The detailed procedure for the preparation of the complex coacervates is shown in the Materials and Methods section. In order to study the effect of PNIPAM content, samples were prepared by mixing aqueous solutions of oppositely charged graft polyelectrolytes having a similar PNIPAM content (e.g., PAA-*g*-PNIPAM30 + PDMAPAA-*g*-PNIPAM30), whose synthesis is explained in detail in the Materials and Methods section. In order to study the effect of ionic strength on the mechanical properties of the complex coacervate phase, the final mixture was prepared at three different sodium chloride (NaCl) concentrations: 0.1 M, 0.5 M and 0.75 M NaCl. These conditions were selected in order to explore an interval of concentrations spanning from physiological conditions (0.1 M NaCl) to a value (0.75 M NaCl) just below the critical salt concentration (CSC, the threshold above which complex coacervation is suppressed [9], around 0.8 M NaCl in this system).

The complex coacervates used in this study are listed in Table 1. The samples are named in the following way: PxSy, where P stands for PNIPAM, x is the molar percentage of PNIPAM, S stands for added salt and y is the molarity used for the preparation of the complex coacervates.

To allow injectability, an adhesive should exhibit a fluid character upon delivery. At the same time, upon debonding, a certain degree of elasticity, which can be achieved through an in situ phase transition, is required [26]. In this section we evaluate the ability of the prepared samples to meet these requirements and, based on these findings, we narrow the selection down for further analysis.

#### 2.1.1. Injectability

The complex viscosities recorded at 20 °C at low frequency (ω = 0.1 rad/s), close to the Newtonian plateau, where the obtained values approach the zero shear viscosity (as shown in Appendix A for the sample P40S0.75), are shown in Figure 2A, while the water content is reported in Figure 2B.

The viscosity decreases abruptly when increasing both the ionic strength and the PNIPAM content. Samples prepared at the highest salt concentration (0.75 M NaCl) and at high PNIPAM content (30%–40%) have low values, of the same order of magnitude as glycerol (1–2 Pa*s), enabling injection through a 22-gauge needle [27].

For most samples, the viscosity is correlated to the polymer concentration within the material: complex coacervates, despite being weakly hydrophobic and phase-separated from water, are known to have a high water content [28]. The addition of PNIPAM, which is a hydrophilic polymer below its LCST, leads to an overall increase in water content (Figure 2B and Appendix A). As a result, the polymer concentration and the viscosity of the material decrease, favouring injectability.

In addition to the PNIPAM content, salt concentration is also known to affect the water content of complex coacervates. Starting from samples prepared at conditions close to the critical salt concentration (CSC) [9], a decrease in ionic strength promotes the formation of macro-ion pairs, resulting in a higher level of physical crosslinking and lower water retention (Figure 2B). As expected, an increase in viscosity is correlated to a decrease in water content with decreasing salt concentrations (Figure 2A), for samples with low PNIPAM content (0%–10%). However, at high PNIPAM content (from 20% to 40%), the viscosity drastically increases upon lowering the ionic strength, in contrast to the water content which is surprisingly constant (Figure 2A,B). 

The increase in viscosity is due both to the formation of stronger electrostatic attractions, which slow down the chain dynamics, and to the slightly higher polymer concentration, which, at fixed water content, increases at the expense of the salt concentration. The constant water content might instead be due to a different distribution of the water among the domains at high PNIPAM content [22]: at high salt concentration, when PNIPAM chains are more prone to dehydration due to the salting-out effect of sodium chloride [29], water may be retained by the polyelectrolyte matrix, while, at lower ionic strength, the hydrophilic PNIPAM may allocate the water released by the complex coacervate, with the average water content within the material being constant.

The same trend is observed in the viscoelastic properties at 20 °C, which can be detected by performing frequency sweeps on complex coacervates. At high salt concentration (S0.75), the complex coacervate shows a liquid-like behaviour (Figure 3A), with the loss modulus (*G’*) higher than the storage modulus (*G”*) over the whole range of frequencies (the electrostatic bonds between the polyelectrolyte chains are not very strong, allowing them to slide along forming transient interactions, with macro-ion pairs acting as sticky points [30]. 

By increasing the PNIPAM content (Figure 3A), the moduli and the relaxation time (*τ*), obtained from the inverse of the crossover frequency (*ω_c_*), decrease (the same trend is observed at lower salt concentrations). This observation is in line with the findings of the water content analysis: The higher the PNIPAM content, the lower the polymer concentration and, consequently, the lower the number of electrostatic interactions per unit volume. As a result, the moduli decrease (*G’* ~ *Nk_B_T*, where *N* is the number of active chain segments per unit volume) as well as the relaxation time, leading to a higher chain mobility and a lower viscosity.

By lowering the ionic strength, a sol-gel transition is observed: the moduli increase progressively and become frequency independent, with *G’* exceeding *G”* (Figure 3B).

#### 2.1.2. Liquid-to-Solid Transition

An adhesive, after application, should be able to resist detachment: a low viscosity fluid flows when stress is applied, without offering any significant resistance [13]. The desired cohesive properties can be obtained by a liquid-to-solid transition, which, in this material, can be triggered by either raising the temperature or by lowering the ionic strength (Figure 4). The complex coacervates prepared at ionic strength below 0.5 M NaCl behave as soft gels already before application and no obvious transition is observed by changing the environmental conditions (Appendix A).

Based on these observations, the following analysis will now focus only on samples prepared at 0.75 M NaCl, a salt concentration below the CSC, yet high enough to render the material a low viscous fluid at room temperature (RT). The setting process is then activated using two different environmental triggers (Figure 5), which will be separately described in the following sections. When performing a *temperature switch*, the solidification of the material is obtained by raising the temperature above the LCST, stimulating the collapse and the aggregation of PNIPAM side chains (Figure 5A). When performing a *salt switch*, the reinforcement of the complex coacervate is obtained by lowering the ionic strength of the surrounding environment, leading to the formation of stronger electrostatic interactions between the polyelectrolyte backbones (Figure 5B).

### 2.2. Temperature Switch

In this section the thermally induced transition is studied using differential scanning calorimetry (DSC), small angle X-ray scattering (SAXS), rheology and probe-tack testing.

#### 2.2.1. Differential Scanning Calorimetry (DSC)

The LCST phase transition of PNIPAM is a well-studied process: During demixing, energy is required to break the hydrogen bonds with water, leading to an endothermic transition, which can be monitored by DSC [31]. A representative thermogram is shown in Figure 6A for P40S0.75.

The negative peak in the thermogram indicates the presence of the phase transition upon heating. The enthalpy of the transition Δ*H^⦵^_trans_* is obtained by integrating the area of the peak. In this case, the peak temperature (*T_heat_*) is observed at 23.5 °C, with Δ*H^⦵^_heat_* equal to 1.8 kJ/mol. When cooling, an exothermic peak appears at a slightly lower temperature (*T_cool_* = 21.8 °C) and with a slightly lower enthalpy (Δ*H^⦵^_cool_* = 1.3 kJ/mol). This hysteresis effect is likely due to the different kinetics in the association and dissociation processes of the PNIPAM chains, which are rate-dependent phenomena [32]. The data obtained for all the samples used in this work are reported in Table 2.

Obviously, in PNIPAM-free complex coacervates no transition is observed. When the PNIPAM content is low, it is possible to detect the endothermic peak, but the sensitivity of the instrument is not high enough to provide reliable data on the transition enthalpy.

The LCST is strongly affected by the ionic strength of the surrounding environment. Sodium chloride, because of a salting-out effect, is known to disrupt the hydration shell around the PNIPAM chains, causing a large decrease in the LCST [29], as observed in Figure 6B. *T_heat_* decreases linearly as a function of [NaCl] (the same is valid for *T_cool_*), as expected: by extrapolating the values to zero ionic strength, a value of 36.0 °C is obtained for *T_heat_*, which is in line with the data reported in literature for PNIPAM in demineralized water [33]. 

The transition enthalpy, (around 1.8 kJ/mol) is less than half of the values generally reported for the dehydration of PNIPAM in water (4–7 kJ/mol) [33,34]: this is likely due to the low molecular weight of the PNIPAM side chains (5.5 kDa), known to affect the transition enthalpy, in line with literature data for chains of similar size (1.3 kJ/mol for 5.4 kDa PNIPAM) [35].

#### 2.2.2. Small Angle X-ray Scattering (SAXS)

Small angle X-ray scattering (SAXS) was employed to detect the different arrangements of the domains at the nanoscale as a function of PNIPAM content and temperature (Figure 7).

The high *q* region (0.5–2 nm^−1^) contains information about the average conformation of the single polymer chains. For all temperatures and systems the same profile is observed, suggesting a similar conformation of the individual chains, independent of temperature. More specifically, this scaling (*I* ≈ *q*^−1.7^) indicates that the polymer chains behave nearly as in a semidilute polyelectrolyte solution, attaining a self-avoiding random walk conformation [36]. In the low *q* region (0.06–0.5 nm^−1^), except for the sample without PNIPAM (Figure 7A), an upturn is detected at high temperatures. This upturn is mainly due to the decreased compatibility between PNIPAM and the complex phase leading to the segregation of PNIPAM into domains with dimensions of tens of nanometres (according to the observed *q*-range). The domain size distribution is likely to be broad since well-defined peaks related to the structure factor are not observed: this is mainly ascribed to the high polydispersity of the polymer chains (see Materials and Methods section). Furthermore, the upturn, at high PNIPAM content (Figure 7C), is detected already at temperatures below the LCST (around 23 °C according to the DSC data), which indicates that large fluctuation of PNIPAM concentration already starts at room temperature prior to dehydration.

#### 2.2.3. Water Content Analysis

The adhesion performance can be strongly affected by changes in volume and water content [37,38]. However, differently from PNIPAM hydrogels which are known to shrink and expel water upon going through the phase transition [39], the samples are found to keep the same volume and the same water content below and above the LCST, at any PNIPAM content (Figure 8).

This is most likely caused by trapping of water in pores within the material, expelled by PNIPAM domains upon the phase transition, leading to the formation of a porous structure without an overall change in volume (Figure 4C) [8,40].

#### 2.2.4. Linear Rheology

Temperature and frequency sweeps recorded at 50 °C are shown in Figure 9.

At high PNIPAM content (P40S0.75), an increase in *G’* of almost three orders of magnitude is observed with a crossover between the moduli upon surpassing the LCST (Figure 9A). However, the onset of the transition, which is not as sharp as that evidenced by DSC, is detected at a higher temperature (around 27 °C) than the LCST (23.5 °C): by contrast with the phase transition, which is immediate, the formation of a network might require additional time and energy, especially considering the low PNIPAM molecular weight.

At 50 °C, the sample shows a gel-like behaviour, with both moduli frequency-independent and *G’* higher than *G”* (Figure 9B). The dynamics of the chains have been markedly slowed down due to the longer lifetime of the interactions between the PNIPAM units [41].

When lowering the PNIPAM content, the increase in moduli is progressively smaller. For P0S0.75 (no PNIPAM) both *G’* and *G”* are almost temperature independent (Figure 9A): both moduli are frequency dependent above the LCST, having features typical of a viscous material, with *G”* exceeding *G’* over the whole range of frequencies (Figure 9B).

#### 2.2.5. Non-linear Rheology

Shear start-up experiments were performed at a fixed shear rate and at 50 °C to evaluate the non-linear mechanical properties above the LCST (Figure 10).

Almost all samples show a profile characteristic of fracture: the stress (*σ*) rises linearly as a function of strain (*ε*) until reaching a maximum. After that, the stress drops quickly, indicating failure of the physical network [42]. The sample without PNIPAM (P0S0.75) does not show a well-defined peak, with the stress increasing slowly as a function of strain, until reaching a steady-state value. This response is typically observed for a viscoelastic liquid, since the material does not fracture but simply flows as an effect of the applied deformation. The slope of the linear part and the area under the curve increase as a function of the PNIPAM content, meaning that the addition of thermoresponsive units not only stiffens the material, but also favours energy dissipation, improving the material properties both in linear and non-linear deformations.

#### 2.2.6. Underwater Adhesion

An underwater probe-tack test was performed on complex coacervates prepared at 0.75 M NaCl at a fixed strain rate (0.2 s^−1^) and at 50 °C. As shown in Figure 11, the higher the PNIPAM content, the better the underwater adhesion performance.

Except for the sample without PNIPAM (a liquid which does not oppose resistance to the applied stress), all stress-strain curves show a peak and a subsequent decay to zero at a strain value between 100% and 200% (Figure 11A). The mode of failure is always cohesive, with residues of material left on the probe at the end of the experiment (as evidenced from the residual stress at high strain when plotting the curves in log-lin scale, Appendix A). The work of adhesion (*W_adh_*), which is the area under the curve normalized by the thickness of the adhesive layer, increases non-linearly as a function of the PNIPAM content (Figure 11B), reaching 3.9 J/m^2^ at the highest PNIPAM content studied (P40S0.75): this value is comparable to the results obtained when measuring other (bioinspired) adhesives tested with a probe-tack technique in wet conditions [43,44]. These measurements confirm that the addition of PNIPAM is beneficial for both the mechanical and adhesive properties of the complex coacervate. However, a further increase in PNIPAM content is not beneficial since for PNIPAM contents above 40% mol/mol complex coacervate formation is suppressed due to an increased affinity for water below the LCST.

### 2.3. Salt Switch

When raising the temperature above the LCST, the reinforcement of the material is due to the collapse of the PNIPAM units. However, at 0.75 M NaCl the electrostatic interactions between the oppositely charged polyelectrolytes are very weak and do not contribute much to the strength of the system. To activate these interactions and probe the strength of the electrostatic bonds, the complex coacervate phase is immersed in a lower ionic strength medium (0.1 M NaCl) so that the salt ions can diffuse out of the material. Basically, the material is reinforced by applying an ionic strength gradient, defined as salt switch, instead of a temperature switch.

As shown in our previous work [23], the transition takes less than one hour, after which the adhesion experiments can be performed. 

#### 2.3.1. Water Content Analysis

By contrast with what we observed after a temperature-triggered transition, a slight increase in water content is observed after the salt switch (Figure 12A).

The polymer concentration increases as a consequence of salt ions diffusing out, as shown in Figure 12B for the sample P30S0.75. Since the total amount of polymer is constant before and after setting, it turns out that the total amount of water is actually lower, with part of the water molecules diffusing out together with the salt ions. However, for a complex coacervate in equilibrium with its dilute phase, much higher water release is expected (upon this change in ionic strength) than what is observed here: most likely some of the water expelled by the polyelectrolyte matrix remains kinetically trapped in pores within the material [45]. As a result, the adhesive turns immediately opaque when put in contact with the aqueous medium (Figure 4B) because of light scattering due to the porous structure (Figure 4C).

#### 2.3.2. Linear Rheology

The liquid-to-solid transition was monitored in a rheometer by following the evolution of the moduli in time during the salt-triggered setting process (Figure 13). 

As evidenced by time sweeps (Figure 13A), the transition is immediate and the moduli abruptly increase, reaching a constant value after 15 min. *G’* is higher than *G”* in the whole range of frequencies, as evidenced by the frequency sweeps (Figure 13B), indicating the formation of a gel-like material. Since the temperature is kept below the LCST, the strengthening mechanism can only be ascribed to the formation of stronger interactions between the oppositely charged backbones. The moduli increase upon decreasing the PNIPAM content in the material, which is due to the higher number of electrostatic interactions between the polyelectrolytes per unit volume when decreasing the amount of PNIPAM moieties.

#### 2.3.3. Underwater Adhesion

After the salt-triggered setting process, performed at 20 °C, the work of adhesion increases as a function of PNIPAM content up to a critical threshold (P30S0.75, *W_adh_* = 6.5 J/m^2^), after which it drops to very low values for P40S0.75 (Figure 14).

As evidenced in the linear rheology section, the moduli decrease as a function of PNIPAM content, i.e., the presence of the thermoresponsive chains swell and soften the material. However, when PNIPAM is absent (P0S0.75), the high bulk elastic energy stored in the material and weak interfacial interactions lead to detachment from the probe at lower strains than for softer and PNIPAM-containing polymers, as shown in the stress-strain curves (Figure 14A). The adhesive P0S0.75 fails without leaving any residues on the probe: this is known as an adhesive mode of failure. By increasing the PNIPAM content (P10S0.75), a shift from an adhesive to a cohesive mode of failure is observed, together with the formation of filaments able to hold some stress at high deformations. The adhesive fails at a much higher strain in this case and, thereby, the work of adhesion increases (Figure 14B). The presence of the PNIPAM chains, despite being inactive (below the LCST), is therefore crucial because it introduces an additional energy dissipation mechanism, which enables the relaxation of the polymer chains at high deformations [46]. Furthermore, PNIPAM may adsorb more strongly to the substrate, promoting a transition from an adhesive to a cohesive mode of failure. The highest work of adhesion (6.5 J/m^2^) is detected at a PNIPAM content of 30%. However, when the PNIPAM content is too high (P40S0.75), the material becomes too weak to resist fracture: consequently, the peak stress, as well as the work of adhesion, suddenly drops to much lower values, with the sample always failing cohesively.

When comparing the mechanical properties obtained in response to different triggers (Figure 15), it can be noticed that in most of the cases, the adhesive properties are higher when the complex coacervate is solidified using an ionic strength gradient.

By taking a closer look at the stress-strain curves, it can be observed that, despite a stress peak of the same order of magnitude, failure occurs at a much higher strain when performing a salt switch (Figure 16). This might be related to the architecture of the polymer chains, which are composed of long polyelectrolyte backbones and short PNIPAM chains: when applying an ionic strength gradient, stronger electrostatic interactions between the high molecular weight polyelectrolytes are activated. To completely disentangle these long backbones, the adhesive needs to be stretched to a much higher extent than when triggering the collapse of the short PNIPAM chains with a temperature switch. Therefore, this leads to a much higher work of adhesion which, at intermediate PNIPAM content, exceeds almost one order of magnitude the one obtained with a temperature switch. This strongly suggest that, consistent with previous work on PNIPAM [47], associations along the backbone are more effective than with side chains.

This phenomenon might also lead to an increase in the amount of energy dissipated in the material, quantified by normalizing, at fixed frequency, the damping factor tan *δ* (the ratio between *G”* and *G’*) by *G’* [46]. This might indeed explain the difference between the highest work of adhesion values recorded for the two switches: the sample P30 (highest work of adhesion when performing a salt switch) shows a tan *δ/G’* 5.5-fold higher than the sample P40 (highest work of adhesion when performing a temperature switch), ultimately leading to an almost two-fold increase in the work of adhesion (Figure 17).

## 3. Materials and Methods

### 3.1. Materials

Poly(acrylic acid) (PAA, analytical standard, *M_n_* = 239 kg/mol, *M_w_* = 1030 kg/mol), poly(*N*-isopropylacrylamide) amine terminated (PNIPAM-NH_2_, average *M_n_* = 5.5 kg/mol), *N*,*N*′-dicyclohexylcarbodiimide (DCC, 99%), acrylic acid (AA, 99%), potassium persulfate (KPS, ≥99%), *N*-methyl-2-pyrrolidone (NMP, anhydrous, 99.5%), sodium chloride (NaCl, ≥99%), 1-(3-dimethylaminopropyl)-3-ethyl-carbodiimide hydrochloride (EDC, ≥98%), *N*-hydroxysulfosuccinimide (NHS, 98%), allylamine (98%), toluene (anhydrous, 99.8%), formic acid (≥95%) and 1,4-dithioerythritol (≥99%) were purchased from Sigma-Aldrich (Zwijndrecht, The Netherlands). Poly(acrylic acid) (PAA, 25% soln. in water, *M_w_* ≈ 50 kg/mol) was purchased from Polysciences (Paris, France). *N*,*N*-Dimethylaminopropyl acrylamide (DMAPAA, 98%) was purchased from ABCR GmbH (Karlsruhe, Germany). Sodium metabisulfite (Na_2_S_2_O_5_, 98%) was purchased from Scharlau (Barcelona, Spain). (3-Mercaptopropyl)trimethoxysilane (95%) was purchased from Alfa Aesar (Kandel, Germany). Methanol (99.9%), tetrahydrofuran (THF, stab./BHT, 99.8%), diethyl ether (stab./BHT AR, 99.5%) and acetonitrile (ACN, AR, 99.8) were purchased from Biosolve (Valkenswaard, The Netherlands). 1.0 M and 0.1 M Sodium hydroxide solutions (NaOH), 1.0 M and 0.1 M hydrochloric acid (HCl) solutions and CertiPUR^®^ (pH 4.0 buffer solution, citric acid/sodium hydroxide/hydrogen chloride) were purchased from Merck Millipore (Amsterdam Zuidoost, The Netherlands). Tetradecane (99%) was purchased from TCI Europe (Zwijndrecht, Belgium). Millipore water was obtained from Milli-Q (Millipore, conductivity: 0.055 mScm^−1^, Amsterdam-Zuidoost, The Netherlands). Silicon wafers were purchased from ACM (Villiers-Saint-Fréderic, France). Polyvinyl acetate glue (ref. L0196, 20 mL) was purchased from 3M (Cergy, France). Cyanoacrylate adhesive (ref. 495) was purchased from Loctite (Boulogne-Billancourt, France). All products were used as received without further purification.

### 3.2. Polymer Synthesis

Poly(acrylic acid)-*g*-poly(*N*-isopropylacrylamide) (PAA-*g*-PNIPAM) (Figure 1A) was synthesized using a “grafting onto” technique according to the method developed by Durand [48]. Poly(*N*,*N*-dimethylaminopropyl acrylamide)*-g*-poly(*N*-isopropylacrylamide) (PDMAPAA-*g*-PNIPAM) (Figure 1B) was synthesized using a “grafting through” technique. First, a poly(*N*-isopropylacrylamide) macromonomer (macroPNIPAM) was synthesized according to the method developed by Petit [49] and subsequently polymerized together with *N*,*N*-dimethylaminopropyl acrylamide (DMAPAA) to obtain the final copolymer. The detailed synthesis protocol can be found in our previous paper [22].

^1^H-nuclear magnetic resonance spectroscopy (^1^H-NMR) measurements were performed in D_2_O on a Bruker Avance III 400 MHz NMR spectrometer in order to estimate the molar ratio between PNIPAM and polyelectrolyte units. *M_n_* of the cationic copolymer was determined by size exclusion chromatography (SEC) on an Agilent Technologies 1260 Infinity II system using a PSS Novema MAX 1000 Å column with an Agilent 1260 RI detector. Samples were run using water as eluent containing 300 mM formic acid at a flow rate of 0.6 mL min^−1^. The calibration was performed using poly(2-vinylpyridine) standards.

PAA-*g*-PNIPAM and PDMAPAA-*g*-PNIPAM with different PNIPAM content were synthesized by varying the feed ratio between thermoresponsive units and polyelectrolyte moieties, see Table 3. Since the *M_n_* of both the backbone and the side chains for PAA-*g*-PNIPAM are known, it is possible to calculate the total *M_n_* of the copolymer and the average number of side chains using the copolymer composition determined by ^1^H-NMR. These calculations take into account the presence of sodium counterions in the grafted polyanions. For PDMAPAA-*g*-PNIPAM, the macromolecular parameters (*M_n_* and the polydispersity index) and the average composition of the polymers were directly calculated from aqueous SEC and NMR experiments. The ^1^H-NMR spectra obtained for the samples PAA-*g*-PNIPAM30 and PDMAPAA-*g*-PNIPAM30 are shown in the Supporting Information (Appendix A). Similar spectra were obtained for the other graft copolymers.

The polydispersity index (PDI) of the anionic copolymers are not determined, but expected to be high because of the high polydispersity of the PAA backbone (PDI 4.3) and of the PNIPAM side chains (PDI 3.2). The high PDI of the cationic copolymers is likely due to the interactions of the polymer with the SEC column and to the free radical polymerization technique, which does not allow control on the molecular weight. The average number of grafts per backbone spans between 0 and 60 for PAA-*g*-PNIPAM, and between 0 and 9 for PDMAPAA-*g*-PNIPAM (Table 3).

### 3.3. Complex Coacervation

Stock solutions of PAA-*g*-PNIPAM and PDMAPAA-*g*-PNIPAM were prepared at a chargeable monomer concentration (PAA/PDMAPAA moles per unit volume) of 0.15 M. The pH of PAA-*g*-PNIPAM solution was adjusted to 7.0 using 0.1 M NaOH and 0.1 M HCl. 5.0 M NaCl was added to the PDMAPAA-*g*-PNIPAM solution to adjust the ionic strength, followed by an adjustment of the pH to 7.0 using 0.1 M NaOH and 0.1 M HCl. Finally, a calculated amount of PAA-*g*-PNIPAM solution was added to the PDMAPAA-g-PNIPAM solution to reach in the final mixture a 0.05 M total chargeable monomer concentration and a 0.5 mixing ratio (ratio between positive charges and all the charges in the system). If needed, a final adjustment to pH 7.0 was performed.

Oppositely charged graft polyelectrolytes having a similar PNIPAM content (e.g., PAA-*g*-PNIPAM30 + PDMAPAA-*g*-PNIPAM30) were used to prepare the samples. In order to study the effect of ionic strength on the mechanical properties of the complex coacervate phase, the final mixture was prepared at three different sodium chloride (NaCl) concentrations: 0.1 M, 0.5 M and 0.75 M NaCl. These conditions were selected in order to explore an interval of concentrations spanning from physiological conditions (0.1 M NaCl) to a value (0.75 M NaCl) just below the critical salt concentration (CSC, the threshold above which complex coacervation is suppressed, around 0.8 M NaCl in this system).

Complex coacervation took place directly after addition of the PAA-*g*-PNIPAM solution. After vigorous shaking, the complex coacervate phase was dispersed throughout the mixture. The mixture was left to equilibrate for 1 day and then centrifuged at 4000× *g* for 1 h. Two clearly separated phases appeared with the complex coacervate phase sedimenting to the bottom of the centrifuge tube. The complex coacervates were stored at 4 °C to preserve them at a temperature well below the LCST.

### 3.4. Optical Microscopy 

Graft copolymer complex coacervates were imaged with a NikonEclipse Ti2 inverted microscope using a Nikon CFI Plan APO Lambda 10×/0.45 objective. The complex coacervate was squeezed between two glass cover slips and then soaked in a 0.1 M NaCl aqueous solution for an hour. After that, the sample was directly placed onto the microscope in order to record images at room temperature. 

### 3.5. Water Content Analysis

The water content of the complex coacervate phase below and above the LCST was determined as follows. Below the LCST, the dilute phase was removed from the tubes containing the samples. After that, a small volume of complex coacervate phase was loaded into an Eppendorf Tube^®^ and weighed on a Mettler Toledo XS205DU analytical balance. The samples were then freeze-dried for four days. To study the water content above the LCST, the Falcon™ tubes containing both the dilute phase and the complex coacervate phase were left in a water bath at 40 °C for four days. After removing the dilute phase, the same weighting and freeze-drying procedures were performed. The water content was determined by the weight difference before and after the freeze-drying process. Two replicas were conducted to ensure data reproducibility.

The water content after the salt-triggered setting process was investigated by thermogravimetric analysis (TGA) using a SDT Q600 from TA instruments. After removing the dilute phase from the Falcon™ tube, the complex coacervate phase was immersed in a lower ionic strength (0.1 M NaCl) aqueous medium for one hour. After that, the material was directly loaded into the sample holder, a platinum pan, at room temperature. The samples were first equilibrated for 15 min at 110 °C. After that, they were submitted to a temperature ramp from 110 °C to 1200 °C at a heating rate equal to 20 °C/min.

### 3.6. Rheology

Rheological measurements were performed on an Anton Paar MCR301 stress-controlled rheometer using a cone-plate geometry (cone diameter 25 mm, cone angle 1°, measurement position 0.05 mm, glass plate). A Peltier element was used to regulate the temperature. The sample loading was performed as follows. The supernatant was taken off from the Falcon™ tube using a Pasteur pipette, ending up with the complex coacervate phase only. This phase was then applied on the rheometer using a Pasteur pipette and contact with the cone was established at the measurement position. When performing a temperature switch, tetradecane was added around the sample, which was covered with a solvent trap with a metal lid to further prevent water evaporation. The temperature was then raised to 50 °C and a waiting time of 15 min was applied before any measurement. When performing a salt switch on the samples prepared at 0.75 M NaCl, the lower ionic strength aqueous medium (0.1 M NaCl) was applied around the sample at 20 °C, with one hour contact time before performing any rheological experiment. Before loading a new sample, the complex coacervate phase together with the dilute phase was centrifuged at 4000× *g* for 15 min.

#### 3.6.1. Linear Rheology

Amplitude sweeps were performed by varying the strain (*ε*) at a fixed angular frequency (*ω*, 1 rad/s) and at fixed temperature (20 °C) in order to determine the linear regime. It was observed that the storage (*G’*) and loss (*G”*) moduli were constant almost over the whole range of strain amplitudes. A fixed strain equal to 0.5% was then used for all the following measurements.

Frequency sweeps were performed either at 20 °C or at 50 °C at a constant strain of 0.5% in a frequency range between 0.1 and 100 rad/s. Temperature sweeps were performed at a fixed frequency of 1 rad/s and at a fixed strain of 0.5% as the temperature was increased from 20 °C to 50 °C at a rate of 0.5 °C min^−1^. Time sweeps were performed, in order to monitor the evolution of the moduli after a salt switch, at a fixed frequency of 1 rad/s, at a fixed strain of 0.5% and at a temperature of 20 °C. Three replicas were conducted to ensure data reproducibility.

#### 3.6.2. Non-Linear Rheology

Non-linear rheology was used to monitor the mechanical properties at high deformations above the LCST. The temperature was raised to 50 °C and an equilibration time of 60 min was applied. After that, shear start-up experiments were performed by shearing the samples prepared at 0.75 M NaCl at constant shear rate (*γ*, 0.1 s^−1^) and by monitoring the evolution of the shear stress (*σ*) as a function of strain (*ε*). Two replicas were conducted to ensure data reproducibility.

### 3.7. Differential Scanning Calorimetry (DSC)

The PNIPAM phase transition was investigated by DSC measurements using a Q200 from TA instruments. After removing the dilute phase from the Falcon™ tube, the coacervate phase (~40 mg) was loaded into a Tzero Pan at room temperature. The samples, together with a reference filled with the same quantity of solvent (0.1 M/0.5 M/0.75 M NaCl solutions), were first equilibrated for 10 min at 15 °C. After that, they were submitted to temperature cycles from 15 °C to 60 °C at a heating/cooling rate equal to 1 °C/min, which is the lowest scan rate above the sensitivity limit.

### 3.8. Small Angle X-ray Scattering (SAXS)

SAXS experiments were performed at the European Synchrotron Radiation Facility (ESRF) in Grenoble, France, at the Dutch-Belgian Beamline (BM26B, DUBBLE). A Pilatus 1M detector, a fixed energy of 12 keV and a detector distance of 2.7 m were used for all experiments, covering a total *q*-range from 0.0665 nm^−1^ to 5.23 nm^−1^. The two-dimensional images were radially averaged around the centre of the primary beam to obtain the isotropic SAXS profiles. The scattering pattern from silver behenate was used for the calibration of the *q*-range. Eltex was used as reference sample for the intensity calibration in absolute units (cm^−1^). The data were normalized to the intensity of the incident beam to correct for primary beam intensity decay. The data were corrected for absorption and background scattering. Two ionization chambers, placed before and after the sample, were utilized for the measurement of the incident and transmitted beams. The background correction was made by subtracting from the total intensity the contribution of density fluctuations evaluated from measuring the blank (0.75 M NaCl solution). In order to obtain the intensity in absolute units, the scattering profiles were divided by a calibration factor, obtained by dividing the intensity recorded for the Eltex scattering peak by its reference value (24.4).

The samples prepared at 0.75 M NaCl were loaded into 2 mm quartz capillaries using Pasteur pipettes and stored at 4 °C until the measurement was performed. Before starting the experiment, the samples were placed in a Linkam DSC 600 oven that allows temperature control. A temperature ramp from 10 °C to 50 °C was performed. SAXS images were recorded every 30 s at a fixed temperature, which was kept constant for an interval ranging from 5 to 20 min depending on the temperature selected. When a new temperature was selected, the heating rate was fixed to 10 °C/min.

### 3.9. Underwater Adhesion

Underwater adhesion properties of the complex coacervates prepared at 0.75 M NaCl were measured using a tack test setup developed by Sudre et al. [50] and mounted on a Instron^®^ 5333 materials testing system with a 10 N load cell. The test consists of making a parallel contact and detachment underwater between a homogeneous layer of the complex coacervate (thickness ≈ 0.5 mm) and a poly(acrylic acid) (PAA) hydrogel thin film (thickness ≈ 200 nm) [51].

The 5 × 5 mm^2^ silicon wafer coated with the PAA hydrogel thin film was glued with a polyvinyl acetate adhesive to a mobile stainless steel probe, which was fixed to the load cell and connected to the Instron machine. The complex coacervate sample was deposited onto a glass slide, which was previously fastened to the bottom of the chamber using plastic screws and aligned with the probe. Contact between the clean PAA thin film and the complex coacervate was performed at 20 °C until a 0.5 mm thickness was reached. 

When performing a temperature switch, a 0.75 M NaCl water solution was poured in the chamber and the setup was covered at the top with a rubber layer providing heat insulation and temperature control. The whole chamber was heated to 50 °C using a temperature control equipment and the probe was kept motionless for 15 min. When performing a salt switch, a 0.1 M NaCl water solution was poured in the chamber at 20 °C and one hour contact time between sample and probe was applied.

Detachment was then performed at a fixed strain rate of 0.2 s^−1^. Raw data of force and displacement were converted into stress and strain values to obtain the work of adhesion. The strain *ε* was obtained by normalizing the displacement by the initial thickness of the sample (*T*_0_). The normalized stress *σ* was obtained by dividing the force by the thin film contact area. The work of adhesion *W_adh_* was then calculated as follows:(1)$$W_{adh}=T_{0}\int_{0}^{\varepsilon_{max}}\sigma d\varepsilon$$

Three replicas were conducted for every experiment to ensure data reproducibility.

### 3.10. Poly(acrylic acid) PAA Hydrogel Thin-Film Synthesis

The PAA hydrogel thin film was synthesized by simultaneously crosslinking and grafting ene-functionalized poly(acrylic acid) (PAA) onto thiol-modified wafers through a thiol−ene click reaction according to the protocol developed by Chollet et al. [51]. The detailed protocol is reported in our previous work [22].

## 4. Conclusions

In this work, we have studied the effect of ionic strength and of PNIPAM content on the properties of thermo- and salt-responsive complex-coacervate based underwater adhesives. Based on the results, we can conclude the following:A high salt concentration, close to the CSC, is necessary to allow injectability of the adhesive;The addition of PNIPAM allows the activation of the setting process via a temperature and/or a ionic strength gradient, resulting in a better performance when compared to PNIPAM-free complex coacervates;When performing a temperature switch, a PNIPAM content of 40% leads to the highest work of adhesion (*W_adh_* = 3.9 J/m^2^);When performing a salt switch, a PNIPAM content of 30% leads to the highest work of adhesion (*W_adh_* = 6.5 J/m^2^).

Because of the good underwater adhesive performance, these findings might illuminate the path towards the development of biocompatible complex-coacervate based adhesives: these could eventually be used as a delivery strategy for injectable glues for biomedical purposes (e.g., soft tissue repair, wound closure).

## 5. Patents

A patent application about part of the work reported in this paper has been filed and is under revision.

## Figures and Tables

**Figure 1 ijms-21-00100-f001:**
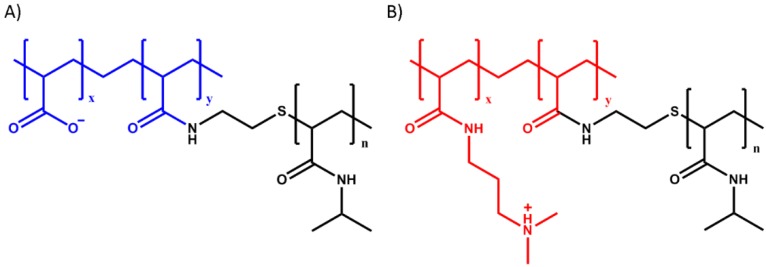
Molecular structure of (**A**) poly(acrylic acid) grafted with poly(*N*-isopropylacrylamide) (PAA-*g*-PNIPAM) and (**B**) poly(*N*,*N*-dimethylaminopropyl acrylamide) grafted with poly(*N*-isopropylacrylamide) (PDMAPAA-*g*-PNIPAM). The coloured parts represent the polyelectrolyte backbones while the black ones represent the PNIPAM units.

**Figure 2 ijms-21-00100-f002:**
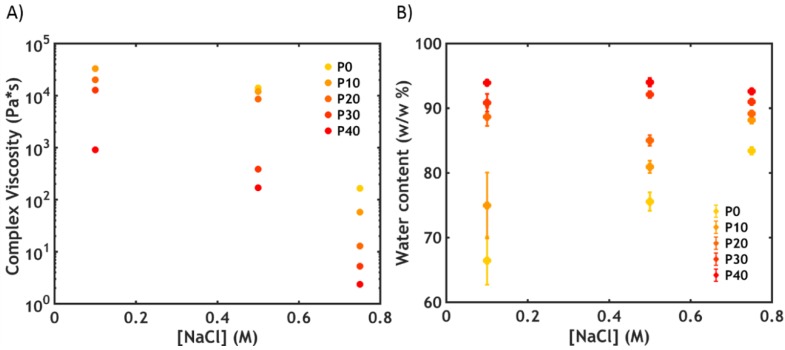
(**A**) Complex viscosity recorded at ω = 0.1 rad/s (P0S0.1 is missing since the measuring position in the rheometer could not be reached because of the high stiffness of the sample) and (**B**) water content at 20 °C as function of [NaCl] in PxSy. Error bars in (**B**) represent standard deviations.

**Figure 3 ijms-21-00100-f003:**
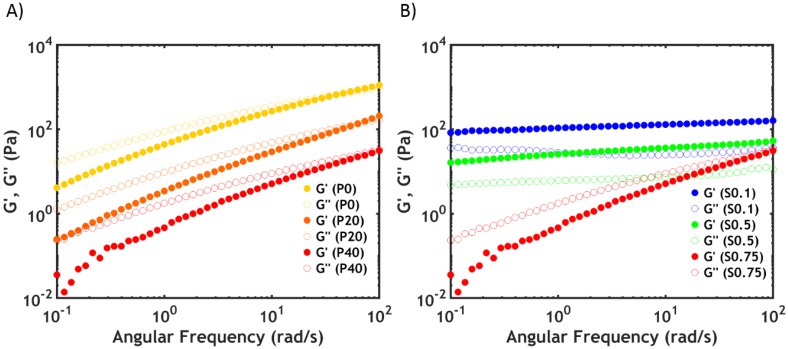
Frequency sweeps performed at 20 °C (**A**) for varying PNIPAM content for PxS0.75 and (**B**) as a function of [NaCl] for P40Sy. Filled dots represent *G’*, while open dots represent *G”*.

**Figure 4 ijms-21-00100-f004:**
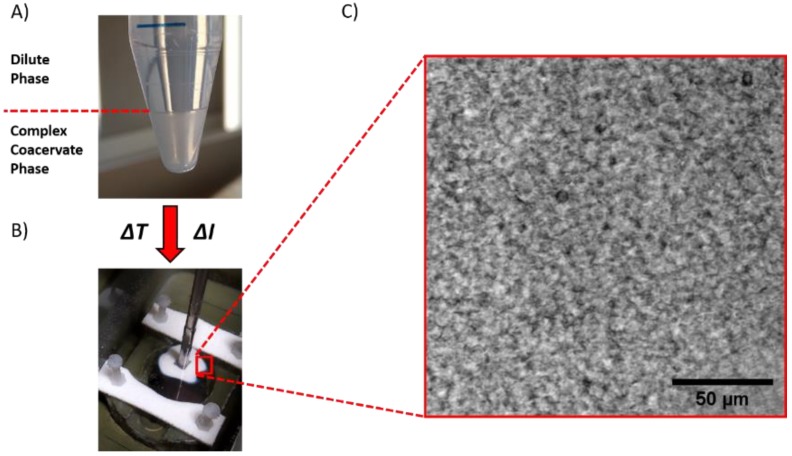
The complex coacervate phase, (**A**) liquid and transparent at low temperature and high ionic strength, turns into a white soft solid when exposed to either a temperature increase (Δ*T*) or (**B**) an ionic strength decrease (Δ*I*). (**C**) The solid material shows a porous structure due to the formation of water pockets.

**Figure 5 ijms-21-00100-f005:**
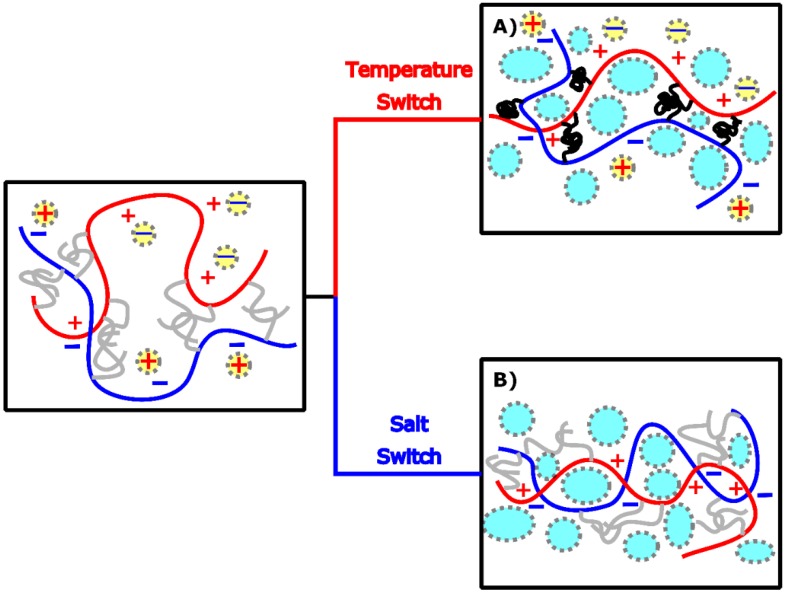
Before setting, the PNIPAM chains (grey) are water-soluble and the electrostatic interactions between the polyelectrolytes (coloured chains) are weak due the high ionic strength (the yellow dots represent the counterions screening the charges). (**A**) When performing a temperature switch, the PNIPAM chains self-assemble in concentrated domains (black domains) releasing water (blue pockets) which remains trapped within the material. (**B**) When performing a salt switch, most counterions are released from the complex and stronger interactions are formed between the polyelectrolytes. The released water remains trapped within the material.

**Figure 6 ijms-21-00100-f006:**
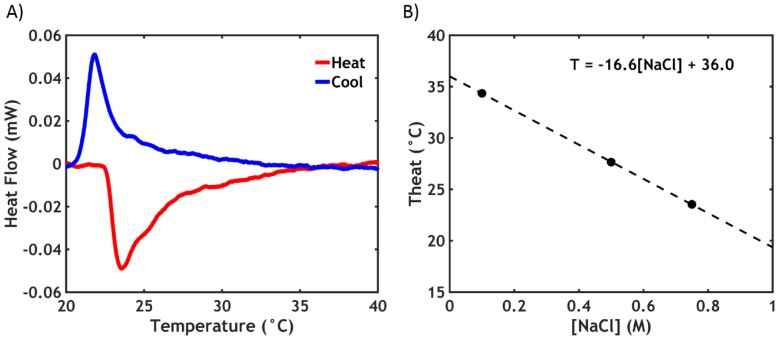
(**A**) Differential scanning calorimetry (DSC) thermogram for P40S0.75 and (**B**) *T_heat_* as a function of [NaCl] in P40Sy.

**Figure 7 ijms-21-00100-f007:**
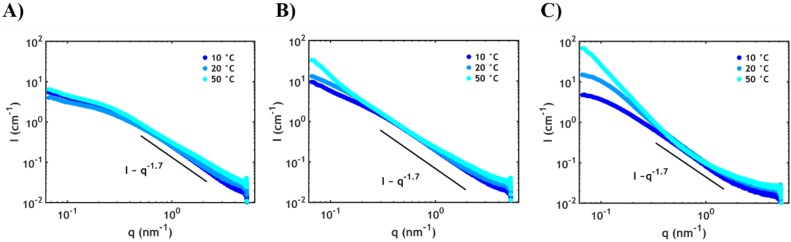
Small angle X-ray scattering (SAXS) plots at different temperatures for (**A**) P0S0.75, (**B**) P20S0.75 and (**C**) P40S0.75.

**Figure 8 ijms-21-00100-f008:**
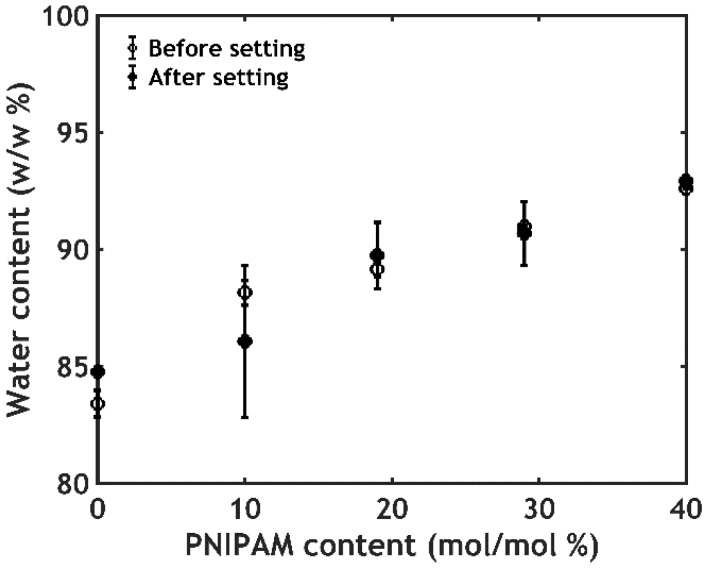
Water content as a function of PNIPAM content for the samples PxS0.75 before and after the temperature-activated setting process. Error bars represent standard deviations.

**Figure 9 ijms-21-00100-f009:**
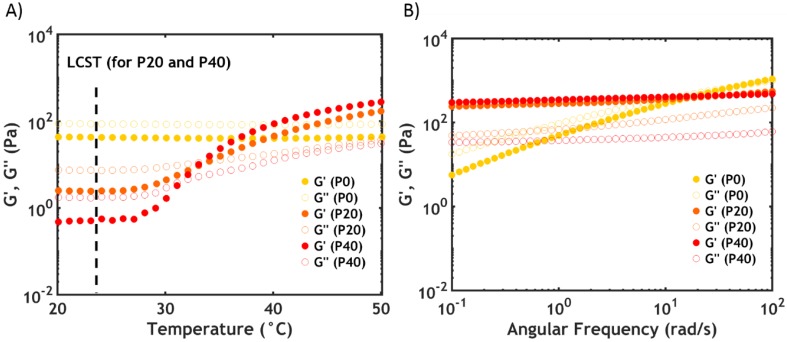
(**A**) Temperature sweeps and (**B**) frequency sweeps performed at 50 °C performed as a function of PNIPAM content for PxS0.75. Filled dots represent *G’*, while hollow dots represent *G”*.

**Figure 10 ijms-21-00100-f010:**
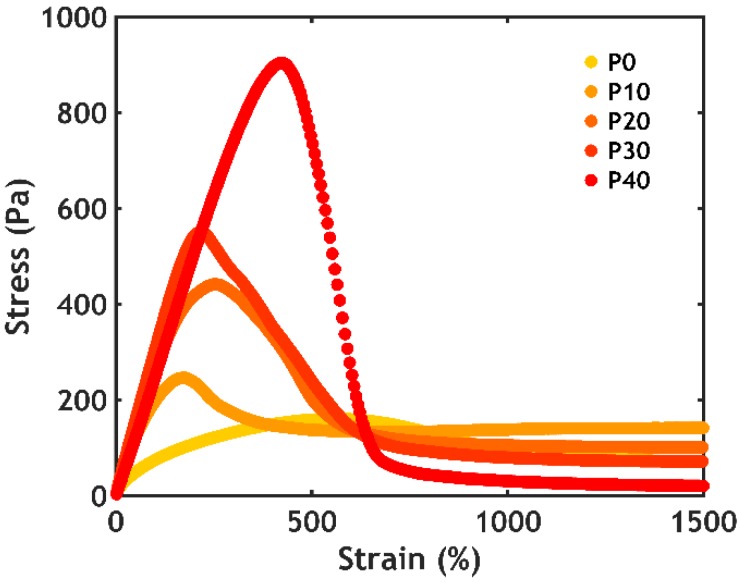
Shear start-up experiments performed at 50 °C as a function of PNIPAM content for the samples PxS0.75.

**Figure 11 ijms-21-00100-f011:**
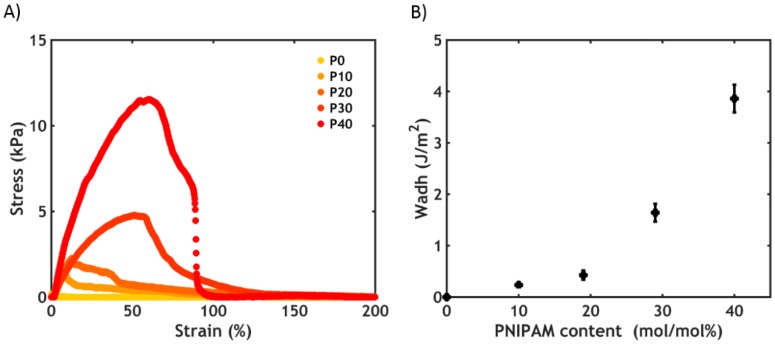
Underwater adhesion performance obtained after the temperature-activated setting process for the samples PxS0.75. (**A**) Stress-strain curves plotted in linear scale and (**B**) work of adhesion as a function of the PNIPAM content.

**Figure 12 ijms-21-00100-f012:**
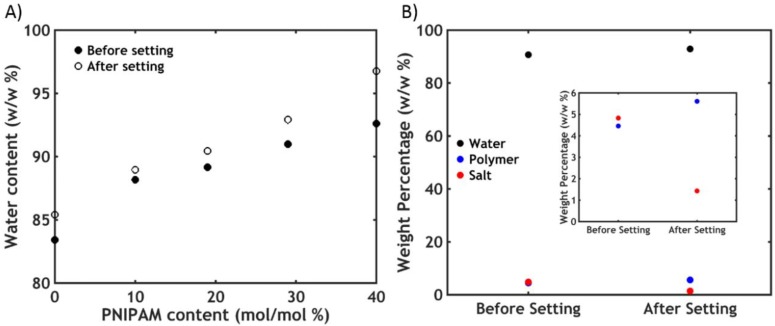
(**A**) Water content as a function of PNIPAM content for the samples PxS0.75 before and after the salt-activated setting process at T = 20 °C. (**B**) Water, polymer and salt content for the sample P30S0.75 before and after the setting process (a magnification of the lower part of the plot is reported in the inset).

**Figure 13 ijms-21-00100-f013:**
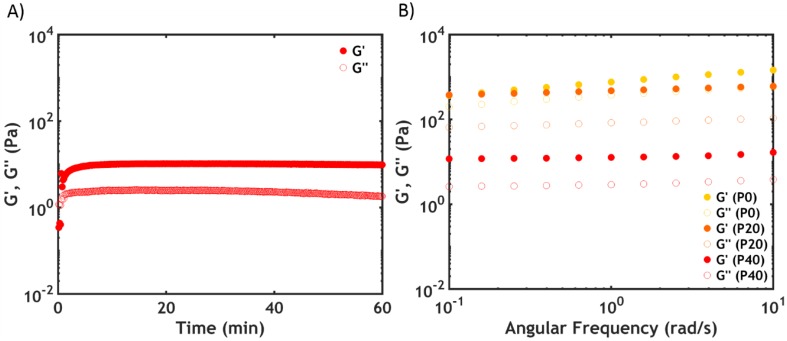
(**A**) Time sweeps performed on P40S0.75 and (**B**) frequency sweeps performed on samples PxS0.75 immediately after the salt-activation at 20 °C. Filled dots represent *G’*, while hollow dots represent *G”*.

**Figure 14 ijms-21-00100-f014:**
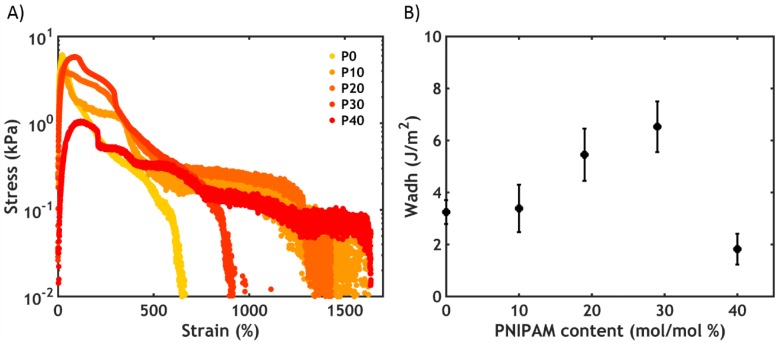
Underwater adhesion performance obtained after the salt-activated setting process for the samples PxS0.75. (**A**) Stress-strain curves plotted in log-lin scale and (**B**) work of adhesion as a function of the PNIPAM content.

**Figure 15 ijms-21-00100-f015:**
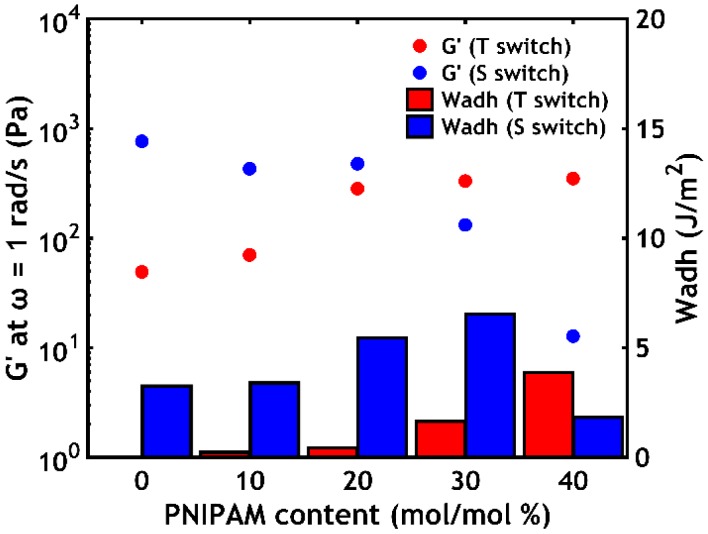
*G’* at *ω* = 1 rad/s and *W_adh_* obtained after a temperature and a salt switch plotted as function of PNIPAM content.

**Figure 16 ijms-21-00100-f016:**
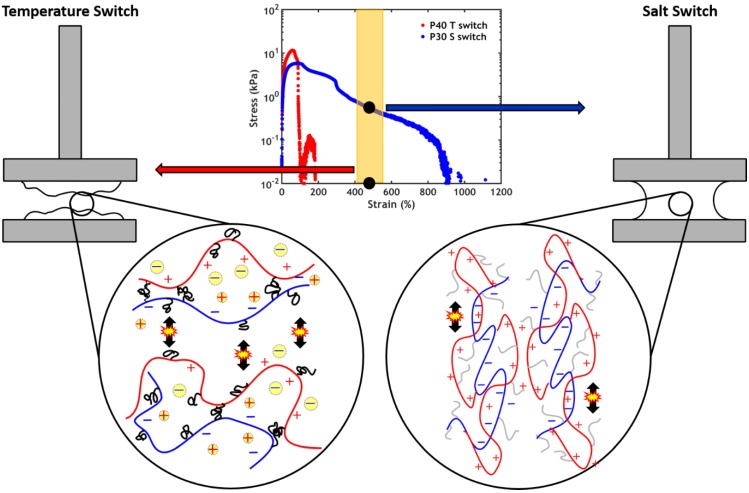
Stress-strain curves for sample P40 (temperature switch) and sample P30 (salt switch): failure occurs at higher strain when applying a salt switch. The architecture of the polymer chains determine the failure process: (left) when performing a temperature switch, the short PNIPAM chains are activated and can be stretched much less than (right) the long polyelectrolyte backbones, whose interactions are activated performing a salt switch.

**Figure 17 ijms-21-00100-f017:**
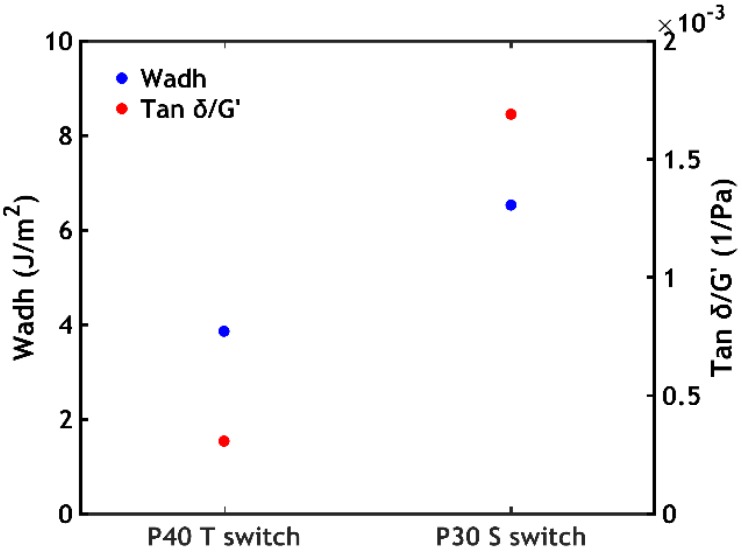
*W_adh_* and tan *δ/G’*(*ω* = 1 rad/s) obtained for sample P40 (temperature switch) and sample P30 (salt switch).

**Table 1 ijms-21-00100-t001:** Complex coacervates analyzed in this work.

Complex Coacervate	PNIPAM/Total Polymer Molar Ratio (% mol/mol)	NaCl Concentration (M)
P0S0.1	0	0.10
P0S0.5	0	0.50
P0S0.75	0	0.75
P10S0.1	10	0.10
P10S0.5	10	0.50
P10S0.75	10	0.75
P20S0.1	19	0.10
P20S0.5	19	0.50
P20S0.75	19	0.75
P30S0.1	29	0.10
P30S0.5	29	0.50
P30S0.75	29	0.75
P40S0.1	40	0.10
P40S0.5	40	0.50
P40S0.75	40	0.75

**Table 2 ijms-21-00100-t002:** DSC data for the analyzed complex coacervates.

Complex Coacervate	*T_heat_* (°C)	Δ*H^⦵^_heat_* (kJ/mol PNIPAM)	*T_cool_* (°C)	Δ*H^⦵^_cool_* (kJ/mol PNIPAM)
P0S0.75	-	-	-	-
P10S0.75	23.8	-	21.6	-
P20S0.75	23.5	-	21.3	-
P30S0.75	22.7	1.8	21.8	1.4
P40S0.75	23.5	1.8	21.8	1.3
P40S0.5	27.7	1.8	26.4	1.3
P40S0.1	34.4	1.9	32.4	1.8

**Table 3 ijms-21-00100-t003:** Graft copolymers synthesized in this work.

Polymer	PNIPAM Molar Ratio (mol/mol %)	*M_n_* Graft Copolymer (kg/mol)	PNIPAM Chains Per Backbone	Polydispersity Index (PDI)
PAA	0	239	0	4.3
PAA-*g*-PNIPAM10	11	359	9	-
PAA-*g*-PNIPAM10	20	405	17	-
PAA-*g*-PNIPAM10	33	499	35	-
PAA-*g*-PNIPAM10	46	636	60	-
PDMAPAA	0	139	0	4.6
PDMAPAA-*g*-PNIPA10	9	104	1	5.3
PDMAPAA-*g*-PNIPA20	18	147	3	6.4
PDMAPAA-*g*-PNIPA30	26	248	7	4.4
PDMAPAA-*g*-PNIPA40	33	244	9	4.2

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
