# Peer review of "Tuning the Interactions in Multiresponsive Complex Coacervate-Based Underwater Adhesives"

_ijms, 2019, doi:10.3390/ijms21010100_

Round 1
Reviewer 1 Report
A well designed study that was well executed and documented. A couple of suggestions would be: 1. Provide the reader with images of the materials at various scales. Also, to strengthen the paper, I recommend adding a graphic or paragraph that illustrates/describes your hypothesis of why you observed that the salt switch with 30% PNIPAM gives an almost double work of adhesion than that of the temperature switch with 40% PNIPAM.
Reviewer 2 Report
I believe the ms is a good ms, with few modifications to do, in particular for conclusions that needs some statments about the future applications of the observation.
But, my main concern is about the suitability of the ms for IJMS. I believe the topic of the ms is out of IJMS aims.
Author Response
We thank the reviewer for their helpful comments regarding the paper. As a result of the comments of the reviewers we have introduced the following changes into the revised version of our manuscript:
GENERAL COMMENTS
I believe the ms is a good ms,
SPECIFIC COMMENTS
with few modifications to do, in particular for conclusions that needs some statments about the future applications of the observation.
We have added the following text and graph to section 4 of the manuscript:
these could eventually be used as a delivery strategy for injectable glues for biomedical purposes (e.g. soft tissue repair, wound closure).
But, my main concern is about the suitability of the ms for IJMS. I believe the topic of the ms is out of IJMS aims.
The aim of the journal, as reported in the website, is the following:
“The International Journal of Molecular Sciences provides an advanced forum for molecular studies in biology and chemistry, with a strong emphasis on molecular biology and molecular medicine.”
We agree with the reviewer that the topic is not closely related to the aim of the journal: however, this paper has been submitted, after an invitation from the editorial board, as part of the special issue "Wet Adhesion: New Chemistries, Models and Translation to Materials". The aim of the special issue, as reported in the website, is the following:
“In this Special Issue, we aim to bring together contributions from different research groups working in the field of underwater adhesion, within a multidisciplinary approach that combines fundamental chemistry and physics, materials science, and engineering. We are interested in contributions that elucidate the mechanisms of bio-adhesion, extract its main design principles as guidelines for development of synthetic adhesive materials, and promote their application to solve current and emerging challenges in aqueous adhesion.”
We therefore believe that our manuscript, focused on the exploration of underwater adhesion properties of a synthetic adhesive material whose design principles are inspired by nature, strongly fits the scope of the special issue.
.
Round 2
Reviewer 2 Report
The authors have performed the requested modifications.